# Non-Structural Carbohydrates and Growth Adaptation Strategies of *Quercus mongolica* Fisch. ex Ledeb. Seedlings under Drought Stress

Yu Wang [1], Xiaoyi Han [1], Wanfeng Ai [2], Hao Zhan [2], Sujuan Ma [1] and Xiujun Lu [1,*]

1   College of Forestry, Shenyang Agricultural University, Shenyang 110866, China
2   College of Horticulture, Shenyang Agricultural University, Shenyang 110866, China
*   Correspondence: lxjsyau@syau.edu.cn

**Abstract:** Under drought stress, plants can change their morphology, physiological characteristics, and carbon allocation to maintain survival and growth. Non-structural carbohydrates (NSC) are major substrates for plant metabolism and play an important role in seedling survival and growth under drought conditions. Mongolian oak (*Quercus mongolica* Fisch. ex Ledeb.), a constructive species distributed in northeast China, has a high drought tolerance. However, studies on seedling growth and the NSC dynamics of Mongolian oak under different drought intensities and durations are limited. To investigate this, our study measured photosynthetic characteristics, growth, biomass, and NSC concentrations for Mongolian oak seedlings on the 0, 15th, 30th, 45th, and 60th day of the experiment under three soil moisture conditions [75% $\pm$ 5% (CK), 50% $\pm$ 5% (W1), and 23% $\pm$ 5% (W2) of soil moisture field capacity (FC)]. Results showed that the growth and biomass gradually decreased as the soil moisture decreased, but the root: shoot ratio and root biomass allocation ratio gradually increased. In the W1 treatment (moderate drought), NSC content in the stems and taproots was 7.42% and 16.39% higher than those in CK at 60 days. However, in W2 treatment (severe drought), NSC content in the stems and taproots was significantly higher than those in CK during the whole period ($p < 0.05$), and they were 14.14% and 26.69% higher than those in CK at 60 days. We found that, under drought stress, Mongolian oak seedlings had lower growth but higher allocation to root biomass and higher NSC content in stems and roots. Furthermore, the root system became a vital carbon sink under drought stress and was beneficial for seedling survival.

**Keywords:** drought stress; biomass allocation; NSC dynamics; *Quercus mongolica*; carbon distribution

## 1. Introduction

Extreme and frequent global drought events substantially decrease tree growth and may lead to tree mortality [1,2]. Seedling survival and growth are susceptible to drought events, which affect forest regeneration, structure, and function [3]; therefore, the physiological mechanisms of seedlings responding to drought has attracted increasing attention. Under drought conditions, water deficit can trigger stomatal closure to prevent water loss. Stomatal closure restricts $CO_2$ from entering the leaf, which leads to a decrease in the photosynthetic rate (Pn) and a reduction of carbon assimilation. At the same time, the morphological and physiological properties of seedlings are modified, which includes the reduction of leaf size and area [4,5], increased leaf mass per area [4,6], root length reduction [7], as well as a reduction in xylem water potential [8], impairment of sap movement [9], and reduced carbon assimilation, leading to a decrease in biomass [5,6].

Non-structural carbohydrates (NSC) are predominantly stored in plants as soluble sugars and starch. As a mobilized carbon resource, NSC not only have storage functions but can also be transformed into compounds with different functions [10,11]. Under drought stress, soluble sugar concentration in plant tissues or cells increases, and then they can directly participate in the osmotic regulation process. Starch serves primarily as storage

and can be converted into soluble sugars [12,13]. NSC assist tree growth, development, defense, and osmotic regulation during drought stress by alleviating the imbalance between photosynthetic supply and metabolic demand, and by exerting their specific protective functions, such as acting as active osmotic compounds and reactive oxygen species (ROS) scavengers [10,13,14].

Different drought durations and intensities can affect NSC allocation and dynamics [12,15,16]. Under drought conditions, stomatal closure reduces C assimilation, and stored NSC are mobilized for metabolic activity. However, during early drought stress, drought can lead to less C demand for growth, and the decline of growth is earlier than photosynthesis, so NSC pools may actually increase [17]. However, during prolonged drought, NSC pools may decrease due to respiratory and metabolic demands [18]. Moreover, some studies have suggested that plants often reduce growth and ensure NSC storage during prolonged drought [19–21]. Previous studies on NSC dynamics have revealed contradictory results [15,20,22], possibly due to different drought durations and intensities. For example, NSC increased under moderate drought [15] but decreased under severe drought in black locust leaves [22]. Patterns of NSC distribution differ among plant organs and are also species-specific [12,23]. In some species, the NSC content of the roots is more sensitive to drought, such as Norwegian spruce [23], but the NSC content in stems is more sensitive to drought in other tree species [22,24]. Currently, some studies primarily focus on one organ or a few organs, but studies on the whole plant level are limited. Therefore, it is necessary to study the NSC dynamics of different organs under different drought intensities and durations in order to better understand the adaptive strategies of trees under drought stress.

Mongolian oak (*Quercus mongolica* Fisch. ex Ledeb.), a deciduous broad-leaved tree species, is widely distributed in northern and northeastern China, Russia, the Korean Peninsula, and Japan [25]. In China, Mongolian oak, as the last ecological defense line for forest ecosystems [26], grows at altitudes of 200–2100 m in secondary forests and has high adaptability to dry environments. However, some studies on the adaptation strategies of Mongolian oak seedlings focused primarily on phenotypic plasticity, photosynthetic physiological characteristics [27], leaf anatomical traits, and chlorophyll fluorescence [26]. The research on NSC distribution and dynamics in different organs under different drought intensities and durations is limited. Therefore, understanding NSC distribution and dynamics is important for interpreting the adaptive strategies of Mongolian oak seedlings under drought stress. In this study, potted manipulation experiments were conducted to analyze the photosynthetic characteristics, growth, biomass, and NSC content of Mongolian oak seedlings under three soil moisture contents [75% ± 5% (CK), 50% ± 5% (W1), and 23% ± 5% (W2) of soil moisture field capacity (FC)] and five different durations (0, 15, 30, 45, and 60 days). We hypothesized that (1) Gas exchange could be restricted to varying degrees, and the growth and biomass of different organs could be inhibited under different drought intensities, (2) NSC content in Mongolian oak seedlings will increase under moderate drought, but decrease under severe drought, and (3) the growth of Mongolian oak seedlings could be limited by carbon storage. Our findings will elucidate the allocation traits of growth and NSC content in different organs of Mongolian oak seedlings and reveal the survival and adaptation strategies of Mongolian oak seedlings under drought stress.

## 2. Materials and Methods

### 2.1. Plant Material and Experimental Design

The experiment was conducted at the experimental base of Shenyang Agricultural University, Shenyang, China (41°49′ N, 123°33′ E). Topsoil (20 cm) was collected and sieved through a 4.75 mm sieve to remove roots and large stones and then placed in plastic pots (32 cm in depth and 29 cm in diameter). According to the bulk density of the sampling area, each pot contained 8 kg of soil, and then the field water holding capacities (FC) and the soil water content were measured using the ring knife method [28]. Mongolian oak seeds were collected from the Liaoning Experimental Forest Farm in Qingyuan County, Fushun, Liaoning Province, and were obtained from a healthy Mongolian oak tree to ensure they

came from the same maternal line. A number of 550 seeds were germinated in wet sand and transferred into plastic pots on 5 May 2021, with one plant per pot. Then, the germinants were grown in a rainout shelter and irrigated 4–5 times per week, 500–600 mL one time. After two months, a total of 360 healthy seedlings with consistent growth (12.5 ± 0.4 cm height and 2.15 ± 0.15 mm basal diameter) were selected for the experiment. During the entire experiment, these seedlings were placed under the rain shelter, which was covered with a transparent plastic film (85% light transmittance) and well-ventilated with open sidewalls. The average temperature of the shed was 25.4 °C, ranging from 17.0 °C to 38.7 °C, and the average humidity was 71.3%, ranging from 35.5% to 93.2% (Figure 1).

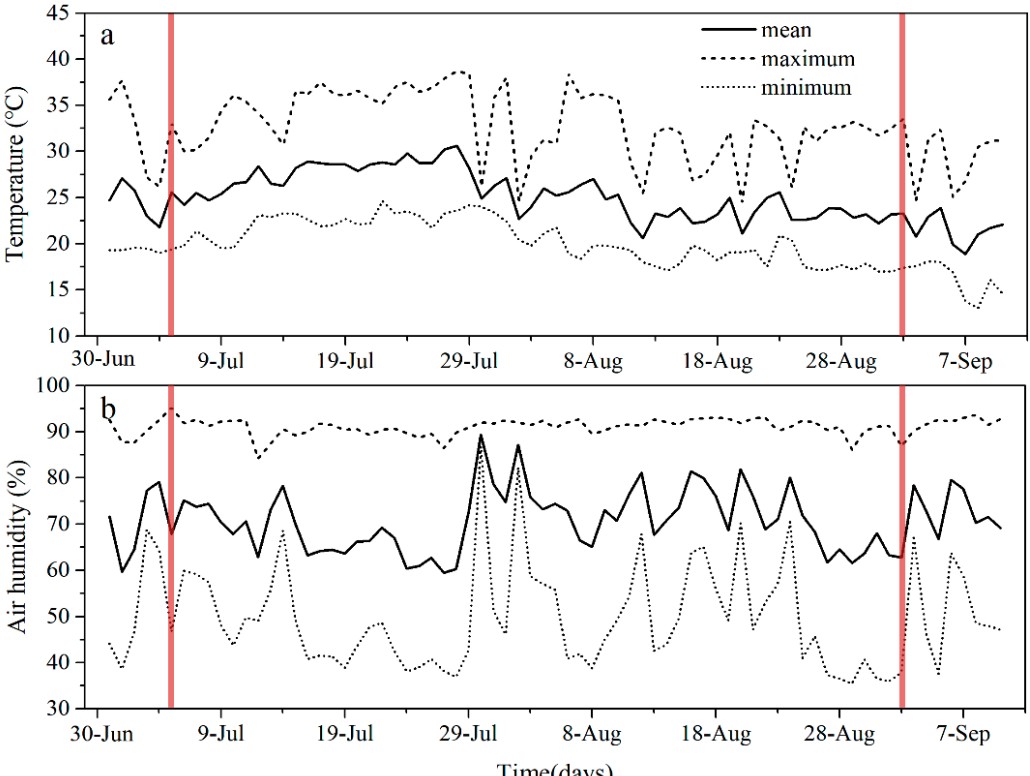

**Figure 1.** Daily mean, maximum, and minimum temperatures (**a**) and air humidity (**b**) measured over time during the whole experiment. The vertical red line shows the beginning and end of the drought period.

The 360 healthy seedlings (two-month-old) were randomly assigned to one of three treatments, which were CK (control group, 75% ± 5%FC), W1 (50% ± 5%FC), and W2 (23% ± 5%FC), with 120 seedlings in each treatment. There were three replications of each treatment, with 40 seedlings in each replication (Figure 2). Potted seedlings were weighed and watered from 17:00 to 19:00 every day to maintain the corresponding soil water content for each treatment. The experiment began on 5 July 2021 and lasted for 60 days. Plant samples were collected on the 0, 15th, 30th, 45th, and 60th day of the experiment to measure leaf gas exchange, growth, biomass, and non-structural carbohydrates.

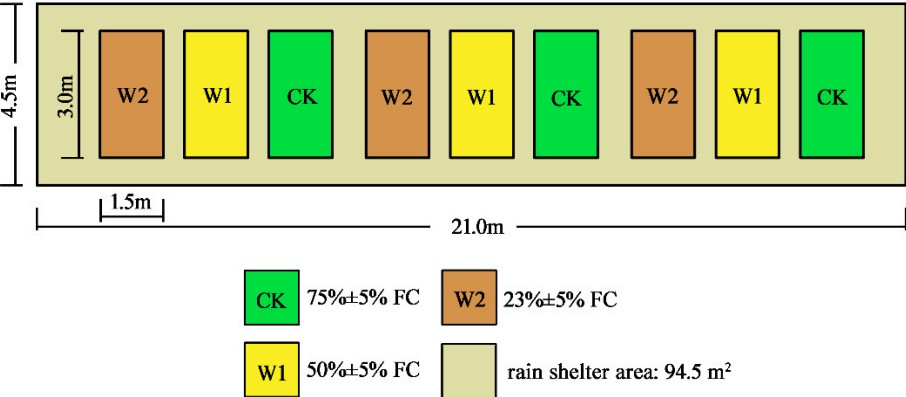

**Figure 2.** The area of the rain shelter and layout of different treatments. FC indicates the field water holding capacities.

### 2.2. Measurement of Leaf Gas Exchange

The photosynthetic rate (Pn) and stomatal conductance (Gs) were measured with an infrared gas analyzer equipped with a red-blue light source (LI-6400-02B, LI-COR Biosciences, Lincoln, NE, USA) on the 0, 15th, 30th, 45th, and 60th day of the experiment. All measurements were conducted at 09:00–11:00 on a sunny day. The light intensity in the leaf chamber was set at 1000 $\mu mol \cdot m^{-2} \cdot s^{-1}$. The ambient $CO_2$ concentration inside the leaf chamber was 400 $\mu mol \, mol^{-1}$ and the air temperature was 20 °C. The relative humidity was approximately 50%.

### 2.3. Measurement of Growth and Biomass

To reflect the growth dynamics, 10 seedlings in each replicate were randomly collected to measure height and basal diameter on the 0, 15th, 30th, 45th, and 60th day of the experiment. Six seedlings were removed from the pot and brought back to the laboratory for cleaning. The total leaf area and root length of three seedlings were determined using an Epson Expression 1680 digital scanner (Epson Electronics Inc., San Jose, CA, USA) and ImageJ analysis software (ImageJ, NIH, Bethesda, Rockville, MD, USA) [29]. The remaining three seedlings were divided into leaves, stems, taproots, and lateral roots, which were oven-dried at 120 °C for 0.5 h and 65 °C for 48 h, and then weighed. The total seedling biomass was the sum of the biomass of all the organs.

### 2.4. Non-Structural Carbohydrates Determination

Dried plant samples (leaves, stems, taproots, and lateral roots) were ground in a ball mill (Scientz-48, Ningbo, China). Soluble sugar concentrations were measured using 0.1 g of dry biomass by extracting soluble sugars in 80% ethanol solution at 80 °C for 30 min and centrifuging them at 5000 rpm for 10 min [8]. The procedure was repeated three times, and the supernatant was collected for sugar analysis using an HPLC 1290 (Agilent 1290 infinity II, Santa Clara, CA, USA). The starch remaining after the ethanol extractions was transformed into soluble substances with perchloric acid, and the concentration of soluble sugar hydrolyzed from starch was determined by the aforementioned method. The starch concentration of each sample was estimated by multiplying the glucose concentration by a conversion factor of 0.9 [30]. The NSC content in each organ was calculated as the sum of the soluble sugar and starch content.

### 2.5. Statistical Analyses

One-way analysis of variance (one-way ANOVA) was used to investigate the effects of varying drought intensities and durations on gas exchange, growth, biomass, and NSC content. Before one-way ANOVA, Levene tests were used to test the homogeneity of the variance. If the variances were homogeneous, mean values were compared using a least significant difference (LSD) multiple comparison test; if the variances were not

homogeneous, the results of the Dunnett T3 method were used. Means were considered significantly different at $p < 0.05$. Statistical analyses of all data were performed using IBM SPSS Statistics 22 (SPSS Inc., Chicago, IL, USA) and Microsoft Excel 2016 (Microsoft Corp., Redmond, WA, USA), and graphs were drawn using the Origin Pro 2015 software (Origin Lab, Northampton, MA, USA).

## 3. Results

### 3.1. Photosynthetic Characteristics and Growth of Mongolian oak Seedlings under Drought Stress

The photosynthetic rate (Pn) and stomatal conductance (Gs) of the Mongolian oak seedlings were significantly affected by drought intensities and durations ($p < 0.05$, Figure 3). The Pn and Gs were low for all treatments at the start of the experiment but increased by day 15 and reached a maximum at day 60 for CK and W1 treatments. In contrast, the Pn and Gs fluctuated with a declining trend over time in the W2 treatment. As a result, the Pn and Gs were 6.84% and 2.36% lower in W1, but 76.64% and 92.77% lower in W2 on day 60, relative to CK seedlings.

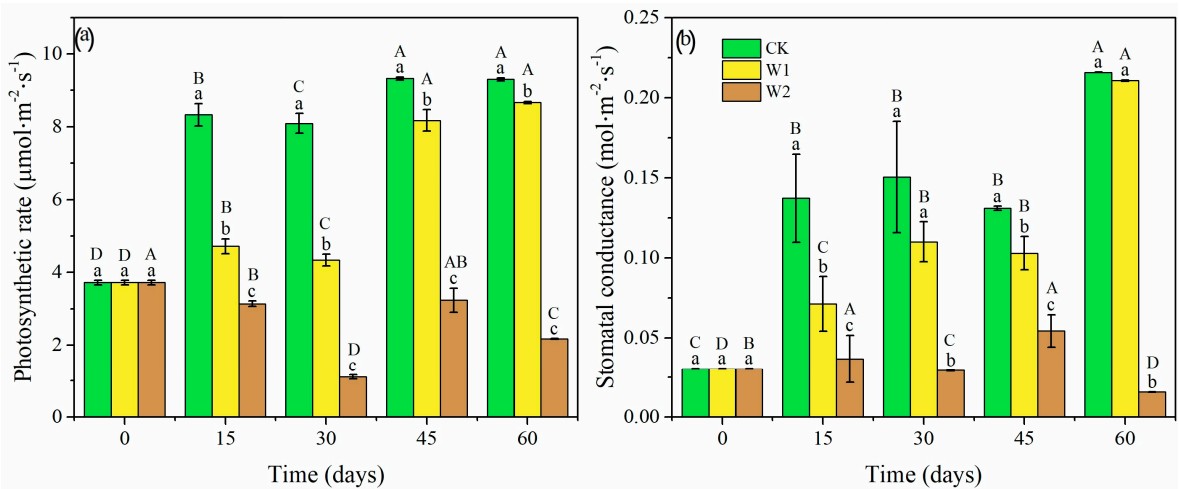

**Figure 3.** Photosynthetic rate (**a**) and stomatal conductance (**b**) in Mongolian oak seedlings under different drought intensities and durations. Uppercase letters represent the significant differences in the same treatment at different times ($p < 0.05$), and lowercase letters represent the significant differences between different treatments at the same time ($p < 0.05$).

Seedling height, stem diameter, total leaf area, and total root length of the Mongolian oak seedlings were significantly affected by drought intensities and durations ($p < 0.05$, Figure 4). Seedlings grew larger in CK throughout the experiment, but seedling height, total leaf area, and total root length were inhibited to varying degrees in the W1 and W2 treatments (Figure 4a,c,d). This was most pronounced for the W2 treatment, as seedling height and total root length did not significantly increase throughout the experiment. Stem diameter was inhibited in W2 during the whole experiment, but was just inhibited at a late stage of W1 treatment (Figure 4b). As a result, seedling height was 16.60% lower in W1 and 30.46% lower in W2 on day 60 (Figure 4a), compared to CK. Stem diameter was 10.64% lower in W1 and 20.85% lower in W2 on day 60 (Figure 4b), relative to CK seedlings. The total leaf areas in W1 and W2 were lower than those in CK (35.50% and 60.34%, respectively) at day 60 (Figure 4c). The total root lengths in W1 and W2 were significantly lower than those in CK (34.65% and 50.64%, respectively) at day 60 (Figure 4d).

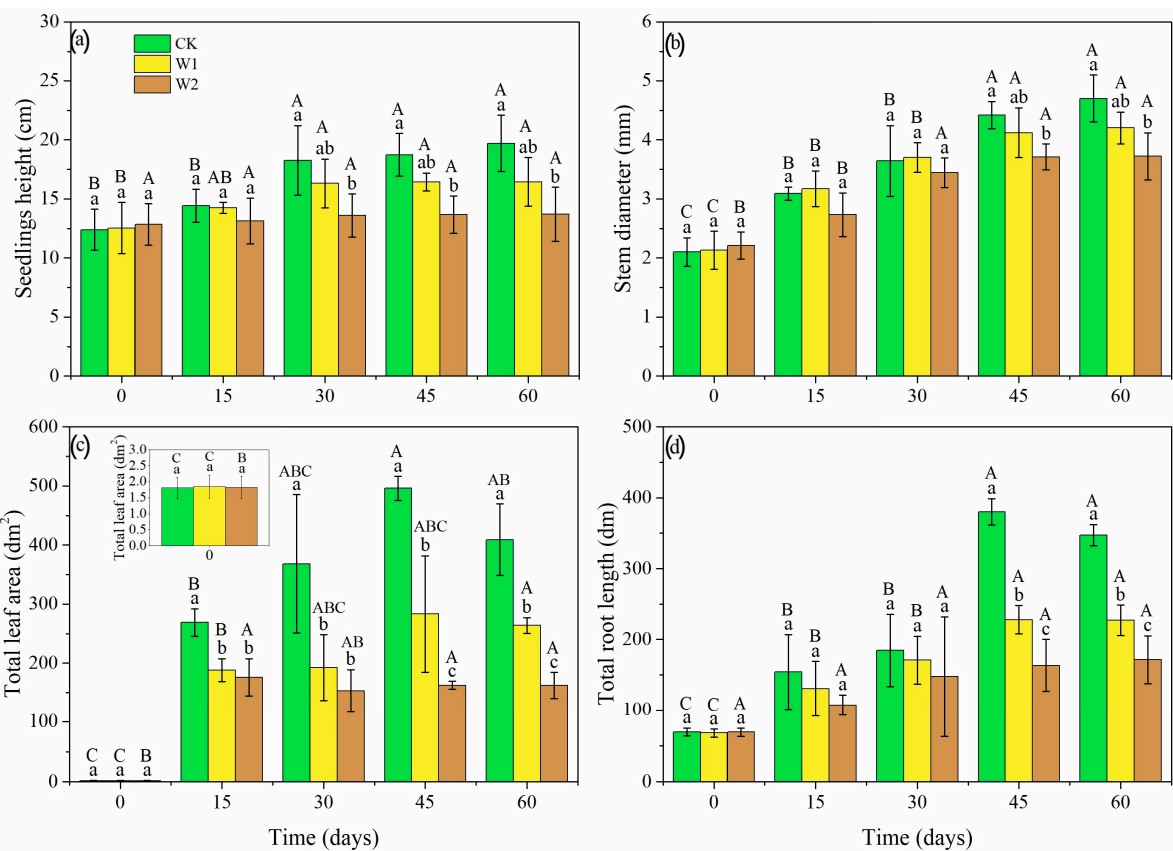

**Figure 4.** Seedling height (**a**), stem diameter (**b**), total leaf area (**c**), and total root length (**d**) of Mongolian oak seedlings under different drought intensities and durations. Uppercase letters represent the significant differences in the same treatment at different times ($p < 0.05$), and lowercase letters represent the significant differences between different treatments at the same time ($p < 0.05$).

### 3.2. Allocation Patterns of Mongolian Oak Seedlings under Drought Stress

There were significant differences in biomass and the root: shoot ratio under different treatments ($p < 0.05$, Figure 5). Biomass of leaves, stems, and roots gradually increased throughout the experiment for CK and W1 treatments, but leaf dry weight did not significantly increase in the W2 treatment (Figure 5a–c). Stem and root dry weights were less inhibited by severe drought and gradually increased from 0 to 45 days. The leaf dry weights were 33.83% lower in W1 and 61.58% lower in W2 at day 60 (Figure 5a), relative to CK seedlings. The stem dry weights were 44.86% and 59.03% lower in W1 and W2 than CK at day 60 (Figure 5b), respectively. The root dry weights were 27.81% and 47.00% lower in W1 and W2 than CK at day 60 (Figure 5c), respectively. The total dry weights were 33.53% lower in W1 and 54.01% lower in W2 at day 60 (Figure 5d), compared to CK. Additionally, drought stress promoted biomass allocation to the roots. With the decrease in soil moisture content and a longer treatment time, root: shoot ratio and the biomass allocation rate of roots gradually increased (Figure 5e,f). The root: shoot ratio in W2 was 45.67% higher than that in CK at day 60. Besides, the biomass allocation rate of roots under W1 and W2 increased by 4.55% and 27.27% at 15 days, 16.67% and 27.08% at 30 days, 1.79% and 10.71% at 45 days, and 5.08% and 11.86% at day 60, compared to CK. Overall, under W1 and W2 treatments, leaf growth was first inhibited followed by stem growth, and then root growth.

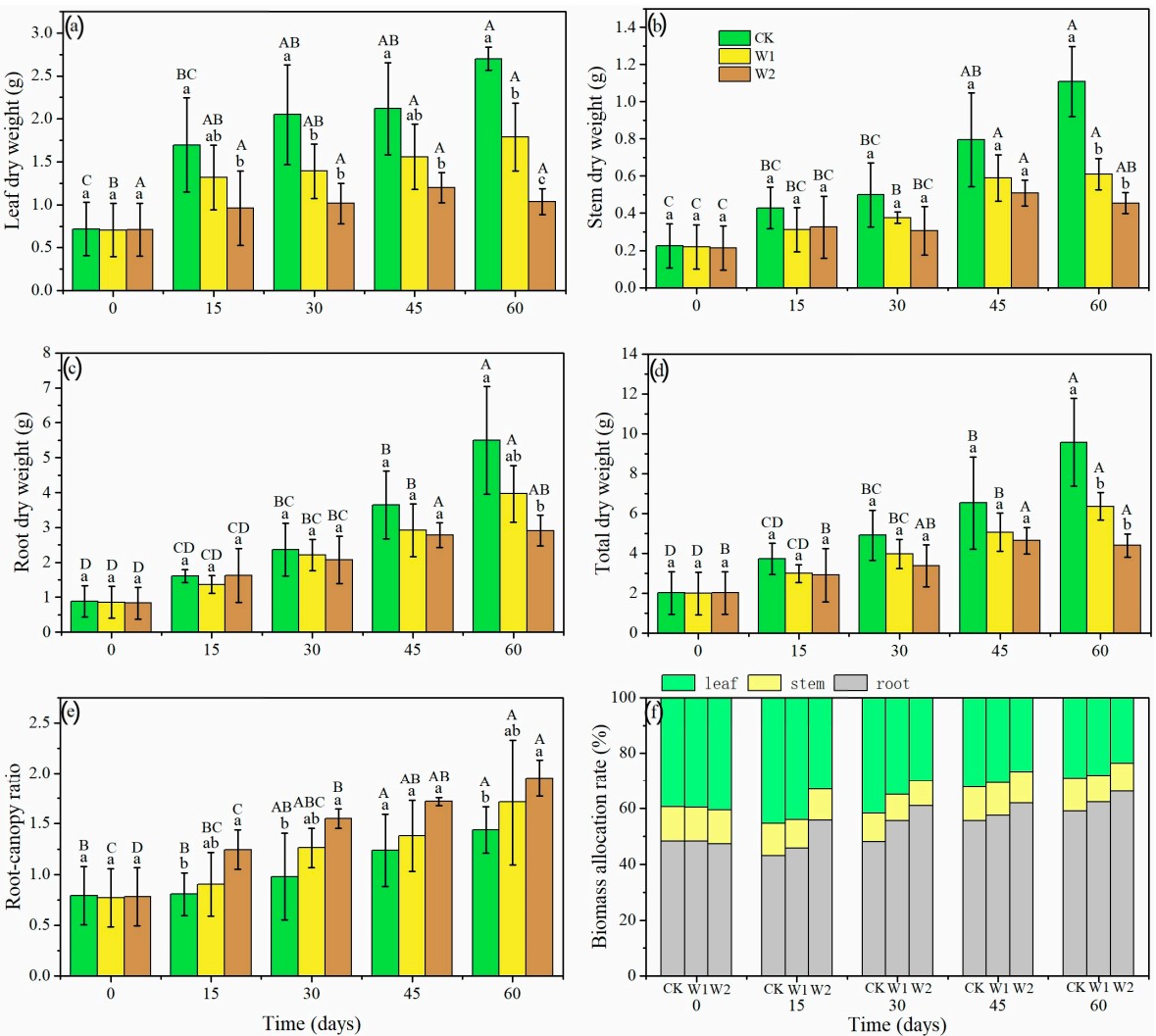

**Figure 5.** Leaf dry weight (**a**), stem dry weight (**b**), root dry weight (**c**), total dry weight (**d**), root: shoot ratio (**e**), and biomass allocation rate (**f**) of Mongolian oak seedlings under different drought intensities and durations. Uppercase letters represent the significant differences in the same treatment at different times ($p < 0.05$), and lowercase letters represent the significant differences between different treatments at the same time ($p < 0.05$).

### 3.3. Dynamics of NSC Content in Different Organs of Mongolian Oak Seedlings under Drought Stress

Drought intensities and durations had significant effects on the soluble sugar content in different organs of Mongolian oak seedlings (Table 1). Soluble sugar content in leaves under W1 and W2 was significantly higher than that under CK at the later stage of treatment, while soluble sugar content in stems in W2 was significantly higher than that under CK at the early stage of treatment. However, the soluble sugar content in taproots and the lateral roots in W1 were significantly lower than those in CK at 30 and 45 days. Besides, taproots soluble sugar content was significantly higher in W2 ($p = 0.023$, 23.12%) at 60 days, and lateral roots soluble sugar content was significantly higher at 15 and 30 days ($p < 0.001$), compared with CK. Thus, under drought stress, soluble sugar content in leaves and stems was higher than that under well-irrigated conditions. However, soluble sugar content in roots was lower in W1 and higher in W2, compared with CK.

**Table 1.** The soluble sugar content of Mongolian oak seedlings in leaves, stems, taproots, and lateral roots under different drought intensities and durations.

| Treatment | Leaf Soluble Sugar | Stem Soluble Sugar | Taproot Soluble Sugar | Lateral Root Soluble Sugar |
|---|---|---|---|---|
| CK (75% ± 5%FC) | | | | |
| 0 | 67.41 ± 0.35 Ba | 36.74 ± 2.43 Ba | 39.81 ± 1.48 Aa | 58.73 ± 0.74 Ba |
| 15 | 51.58 ± 2.91 Da | 34.25 ± 1.36 BCc | 31.01 ± 3.97 Ba | 39.72 ± 1.48 Cb |
| 30 | 62.25 ± 1.93 Ca | 32.79 ± 1.44 Cb | 31.00 ± 0.77 Ba | 60.28 ± 1.99 Bb |
| 45 | 60.61 ± 1.97 Cb | 41.43 ± 0.96 Aab | 41.54 ± 3.71 Aa | 57.33 ± 0.52 Ba |
| 60 | 100.59 ± 3.32 Ab | 40.59 ± 1.43 Aa | 21.80 ± 2.75 Cb | 58.40 ± 3.83 ABab |
| W1 (50% ± 5%FC) | | | | |
| 0 | 67.41 ± 0.35 Ba | 36.74 ± 2.43 Aa | 39.81 ± 1.48 Aa | 58.73 ± 0.74 Aa |
| 15 | 56.00 ± 2.09 Ca | 37.11 ± 1.60 Abc | 33.96 ± 1.33 Ba | 46.70 ± 2.24 Ca |
| 30 | 58.94 ± 4.82 Ca | 37.38 ± 2.78 Aab | 25.63 ± 2.50 Cb | 51.73 ± 2.31 Bc |
| 45 | 73.05 ± 2.76 Ba | 36.60 ± 2.69 Ab | 26.90 ± 3.73 Cc | 22.97 ± 1.04 Dc |
| 60 | 86.82 ± 6.15 Ac | 40.15 ± 3.51 Aa | 20.56 ± 1.31 Db | 56.19 ± 1.34 Ab |
| W2 (23% ± 5%FC) | | | | |
| 0 | 67.41 ± 0.35 Ba | 36.74 ± 2.43 Ba | 39.81 ± 1.48 Aa | 58.73 ± 0.74 Ca |
| 15 | 55.33 ± 1.95 Ca | 42.38 ± 1.70 Aa | 33.73 ± 1.67 Ba | 46.46 ± 3.99 Da |
| 30 | 59.96 ± 0.93 Ca | 40.85 ± 2.76 ABa | 29.92 ± 1.48 Ca | 71.64 ± 2.81 Aa |
| 45 | 72.01 ± 5.80 Ba | 42.14 ± 4.07 Aa | 35.99 ± 1.25 Bb | 57.05 ± 2.06 Ca |
| 60 | 113.28 ± 0.39 Aa | 42.49 ± 2.28 Aa | 26.84 ± 1.78 Da | 63.95 ± 1.82 Ba |

Uppercase letters represent the significant differences in the same treatment at different times ($p < 0.05$), and lowercase letters represent the significant differences among different treatments at the same time ($p < 0.05$).

Drought intensities and durations had significant effects on the starch content in different organs of Mongolian oak seedlings (Table 2). Starch content in leaves was significantly lower in W1 at 30 and 60 days, and it was significantly lower in W2 at 60 days, compared to CK. The stem's starch content in W1 and W2 was significantly higher by 8.77% and 16.13% at 45 days ($p = 0.015$, $p = 0.002$), 12.07% and 19.31% at 60 days ($p = 0.005$, $p = 0.000$), compared to those under CK. The starch content in the taproot under W1 was significantly higher at 60 days ($p = 0.004$), but those under W2 were significantly higher from 15 days to 60 days ($p = 0.001$, $p = 0.000$, $p = 0.000$, $p = 0.004$), compared with CK. There was no significant difference in the starch content in lateral roots between W1 and CK treatment, while it was significantly higher under W2 than that under CK at the later stage of treatment.

**Table 2.** The starch content of Mongolian oak seedlings in leaves, stems, taproots, and lateral roots under different drought intensities and durations.

| Treatment | Leaf Starch | Stem Starch | Taproot Starch | Lateral Root Starch |
|---|---|---|---|---|
| CK (75% ± 5%FC) | | | | |
| 0 | 139.77 ± 0.35 Aa | 42.17 ± 2.43 CA | 174.23 ± 3.10 Aa | 21.74 ± 1.64 Ba |
| 15 | 93.88 ± 6.74 Bab | 34.56 ± 1.10 DB | 122.50 ± 5.25 Cb | 18.58 ± 1.44 Ca |
| 30 | 107.22 ± 0.67 ABa | 52.92 ± 5.80 ABCDA | 166.98 ± 0.60 Ab | 29.23 ± 0.57 Aa |
| 45 | 115.97 ± 3.92 ABab | 56.87 ± 2.50 BC | 153.79 ± 5.64 Bb | 22.64 ± 2.09 Bb |
| 60 | 164.95 ± 2.04 ABa | 74.36 ± 2.15 AB | 116.37 ± 4.38 Cc | 26.61 ± 2.24 Ab |
| W1 (50% ± 5%FC) | | | | |
| 0 | 139.77 ± 0.35 Aa | 42.17 ± 2.43 DA | 174.23 ± 3.10 Aa | 21.74 ± 1.64 Ba |
| 15 | 92.46 ± 1.28 Cb | 42.10 ± 2.51 DAB | 99.79 ± 2.74 Cb | 19.48 ± 0.22 Ca |
| 30 | 90.55 ± 3.71 Cb | 47.14 ± 3.44 CA | 179.16 ± 3.79 Aab | 27.05 ± 1.96 Aab |
| 45 | 110.39 ± 7.46 Bb | 61.86 ± 1.60 BB | 153.89 ± 4.08 Bb | 21.43 ± 1.41 BCb |
| 60 | 133.63 ± 4.32 Ab | 83.34 ± 3.45 AA | 140.91 ± 3.97 ABCb | 27.96 ± 0.14 Aab |

Uppercase letters represent the significant differences in the same treatment at different times ($p < 0.05$), and lowercase letters represent the significant differences among different treatments at the same time ($p < 0.05$).

**Table 2.** *Cont.*

| Treatment | Leaf Starch | Stem Starch | Taproot Starch | Lateral Root Starch |
|---|---|---|---|---|
| W2 (23% ± 5%FC) | | | | |
| 0 | 139.77 ± 0.35 Aa | 42.17 ± 2.43 DA | 174.23 ± 3.10 Aa | 21.74 ± 1.64 CDa |
| 15 | 101.87 ± 1.77 Cab | 44.42 ± 0.63 DA | 153.04 ± 8.72 ABa | 19.75 ± 1.73 Da |
| 30 | 112.16 ± 8.99 Ca | 50.50 ± 3.81 CA | 182.48 ± 0.85 Aa | 24.29 ± 2.28 BCb |
| 45 | 118.73 ± 2.31 Ba | 66.05 ± 1.45 BA | 182.14 ± 0.76 Aa | 32.69 ± 2.64 Aa |
| 60 | 67.61 ± 7.21 Dc | 88.73 ± 1.45 AA | 148.20 ± 2.56 Ba | 30.02 ± 1.60 ABa |

Uppercase letters represent the significant differences in the same treatment at different times ($p < 0.05$), and lowercase letters represent the significant differences among different treatments at the same time ($p < 0.05$).

Drought intensities and durations had significant effects on NSC content in different organs of Mongolian oak seedlings ($p < 0.05$, Figure 6). The NSC content in leaves under W1 and W2 was 16.98% and 31.88% lower than those in CK at 60 days (Figure 6a). However, the NSC content in stems was significantly higher under W1 and W2 than those under CK at 15 and 60 days, and the NSC content in stems gradually increased over time (Figure 6b). The NSC content in taproots was lower under W1 at 15 and 45 days, but higher at 60 days ($p = 0.002$), compared to CK (Figure 6c). However, the NSC content in taproots was significantly higher under W2 than that under CK from 15 days to 60 days ($p = 0.003$, $p = 0.000$, $p = 0.002$, $p = 0.000$). At 60 days, the NSC content in taproots under W1 and W2 was 16.39% and 26.69% higher than those in CK. Besides, the NSC content in lateral roots was lower under W1 at 30 and 45 days ($p = 0.000$, $p = 0.000$, Figure 6d), but it was higher under W2 at 15 and 30 days, compared with CK ($p = 0.022$, $p = 0.000$).

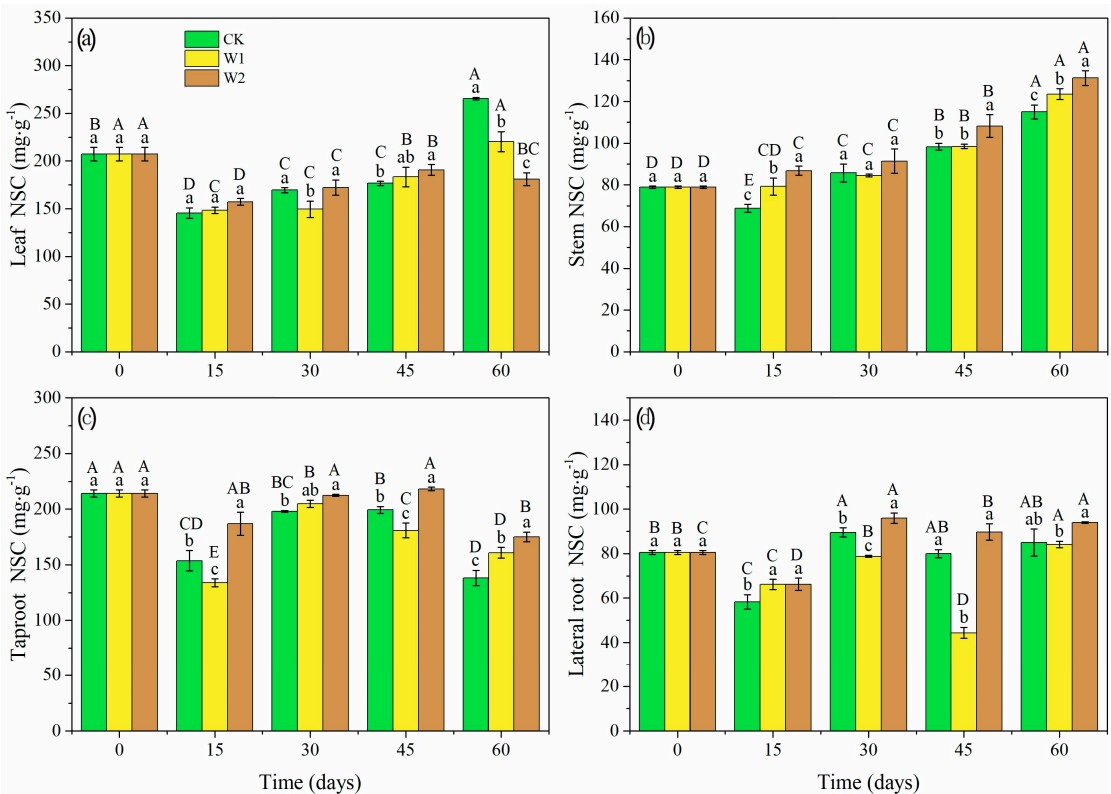

**Figure 6.** Leaf NSC content (**a**), stem NSC content (**b**), taproot NSC content (**c**) and lateral root NSC content (**d**) of Mongolian oak seedlings under different drought intensities and durations. Uppercase letters represent the significant differences in the same treatment at different times ($p < 0.05$), and lowercase letters represent the significant differences between different treatments at the same time ($p < 0.05$).

*3.4. Trait Correlation*

Correlation analyses showed that stem diameter, root and stem dry weights, total dry weights, and the root: shoot ratio of the Mongolian oak seedlings were positively correlated with the soluble sugar content in leaves, starch and NSC content in stems under drought stress (Figure 7). Total root length was positively correlated with starch and NSC content in the stem. Root dry weight was also positively correlated with NSC content in leaves. However, seedling growth and biomass were not related to NSC content in taproots.

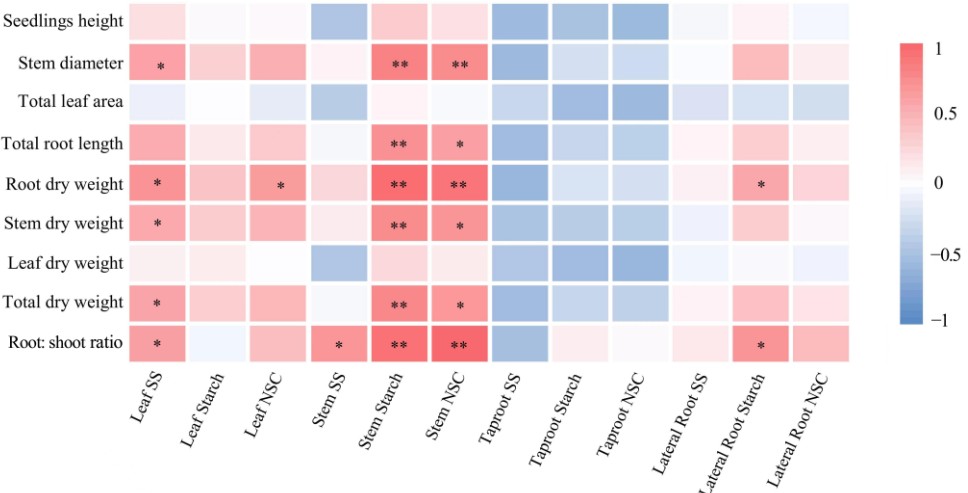

**Figure 7.** Correlation coefficient matrix of Mongolian oak seedlings growth, biomass, and NSC content in each organ. * $p < 0.05$; ** $p < 0.01$.

## 4. Discussion

Our results showed that the Pn and Gs of Mongolian oak seedlings were constrained under drought, which was most pronounced under severe drought. With the decrease in soil moisture content, there was a gradual decline in leaf, stem, and root growth, but the root: shoot ratio and root biomass allocation ratios steadily increased. Besides, under moderate drought, the total NSC content in roots decreased significantly, mainly due to the significant reduction of sugar content, however, it was higher than that under normal water conditions at the end of treatment. Under severe drought, the NSC content in taproots was higher than that under well-watered treatment during the whole experiment. The results did not support our second hypothesis. In addition, the total NSC content in leaves decreased at the end of treatment, while those in stems steadily increased under moderate and severe drought. Moreover, in contrast to our third hypothesis, the reduction of seedling growth was not accompanied by the decrease in the starch content in different organs under drought stress, indicating the decline in growth was not limited by carbon storage.

*4.1. Effects of Drought Stress on Photosynthetic Characteristics, Growth, and Biomass*

In this study, we observed that the Pn and Gs of Mongolian oak seedlings were significantly constrained under drought conditions. The Pn was 6.84% lower in W1 and 76.64% lower in W2, while the Gs was 2.36% lower in W1 and 92.77% lower in W2 on day 60, compared to CK. This has been confirmed by many studies. Drought can cause stomatal closure, limit the entry of carbon dioxide, and further affect the photosynthetic rate [31,32]. Moreover, there was a synergistic change between the Pn and Gs [33]. Therefore, the reduction in photosynthesis under drought is likely due to stomatal limitation, which was more pronounced under severe drought stress. Reduction of carbon uptake for prolonged periods of severe drought may lead to a reduced photosynthetic capacity of mesophyll cells [26,34], reduced chlorophyll content [4,31], and decreased leaf area [4,6]. Drought can also impair sap movement, affecting carbon transport among plant organs [5,31].

Growth, the result of cell division and expansion, is the process that is most sensitive to drought [17]. In this study, we observed that the growth of Mongolian oak seedlings was inhibited, and seedling height and total leaf areas even stopped growing in the early stage of severe drought. Other studies have also found that stem elongation growth, secondary growth, and leaf areas of seedlings are significantly lower under drought stress than those under non-stressed conditions [35,36]. This may be because water deficits strongly reduce leaf or shoot expansion rates [37]. Additionally, our results showed that the total biomass and biomass of each organ were restrained, which was demonstrated in Persian oak under different drought intensities [6,38]. However, the root: shoot ratio increased under drought conditions, and it was 45.67% higher under severe drought than that under normal conditions. These results indicate that plants adjust the distribution of photosynthates among organs, and distribute more products from the source organs (leaves) to the sink organs (root) under drought stress [39]. Root growth is beneficial for obtaining more water and nutrients. Overall, regulation of growth and biomass allocation reduces water loss and promotes survival under drought conditions [40].

### 4.2. Effects of Drought Stress on NSC Distribution of Mongolian Oak Seedlings

Soluble sugars are organic solutes that promote osmotic regulation in higher plants under drought stress [41,42], and they are crucial to osmotic regulation and transport in the vascular system. Our results showed that the soluble sugar content in the stems and lateral roots was higher at the beginning of drought stress than in non-stressed conditions. Silva et al. [41] observed that the total soluble sugar content positively increased with a more severe water deficit and showed the largest relative contribution to osmotic adjustment. An increase in soluble sugar concentration can reduce water potential, maintain turgor pressure, enhance root water absorption capacity [43], and maintain normal physiological activities to acclimate to drought stress [44].

As time progressed, we observed that the NSC content was 12.87% lower in taproots and 11.99% lower in lateral roots under moderate drought than those under non-stressed conditions, along with the decrease in starch and NSC content in leaves. Under sustained moderate drought stress, the Pn was restricted for a considerable time and carbon assimilation decreased [45]. To meet the needs of root growth and metabolism, the soluble sugar in roots was consumed and reduced. This decrease in the NSC content in leaves might be due to the redistribution of NSC from leaves to roots under drought stress [46,47]. However, under severe drought conditions, we observed that the starch and NSC content was 28.53% and 26.13% higher in stems and 24.93% and 21.67% higher in taproots than those under non-stressed conditions. This might be because the growth of seedlings and synthesis of structural substances were greatly limited, which led to a reduction in the use of photosynthates. When the growth rate decreases faster than the Pn, more photosynthates are stored as NSCs [17]. Galvez et al. [20] also observed the increase of soluble sugar and starch content in the roots and leaves of aspen (*Populus tremuloides* Michx.) seedlings under severe water stress. It has been suggested that using NSC to maintain osmotic pressure and hydraulic function was the basis for improving drought resistance, and higher NSC concentrations could delay drought mortality [48]. In a stressed environment, trees actively reduce growth rates and increase carbon storage to improve survival rates [49]. When carbon assimilation is reduced, plants adjust the allocation of carbon between storage and growth to prevent the risk of long-term carbon starvation [43,50].

At the end of this experiment, which was also the end of the growing season, the starch content in leaves was significantly decreased under moderate and severe drought conditions, but soluble sugars in leaves were higher than in normal conditions. This might be because stored starches are converted into soluble sugars to meet the needs of osmotic adjustment, which is an important drought response [12,18]. The starch and NSC content in stems and taproots were also significantly higher, compared to those under non-stressed conditions. Sanz-Pérez et al. [51] observed that oak plants under drought stress would replenish their carbon storage in autumn and allocate more biomass

and starch reserves to their roots, which is consistent with our results. Under stress conditions, plants strategically allocate carbohydrate resources to acclimate to adverse environmental conditions. Carbohydrate accumulation in roots was closely related to growth and physiological changes. At the end of the growing season, the roots preferentially perform the reserve function and accelerate carbon storage while stopping growth, which may be an adaptive strategy for Mongolian oak seedlings under drought stress.

*4.3. Trade-Off between Seedling Growth and Carbon Storage under Drought Stress*

There is no consensus on the trade-off between tree growth and carbon storage. Some studies suggested that plant growth under drought stress is not restricted by limited photoassimilates [17,52]. Cell growth or organ expansion was affected earlier than photosynthesis, so carbon storage increased [17]. Klein et al. [35] observed that different drought stress levels had no significant effects on starch and soluble sugar concentrations in branches, although the stem growth and needle length were reduced. However, at specific developmental or phenological stages or under extreme circumstances, such as persistent heavy defoliation, growth can be limited by carbon availability [53]. Thus, there may be a trade-off between growth and storage [42]. In this study, drought stress inhibited the photosynthesis of Mongolian oak seedlings and also limited their growth. However, the reduction of seedling growth was not accompanied by the decrease in the starch content in different organs. Besides, under severe drought and at the end of moderate drought, starch and NSC content in roots and stems were significantly higher than those under non-stressed conditions. Therefore, root growth of Mongolian oak seedlings is likely not limited by carbon availability, at least for the first two months of the drought. Galvez et al. [20] also observed that limited photosynthates were preferentially stored in the form of starch under severe water stress in aspen seedlings, while growth was restricted. Piper et al. [19] also observed that starch concentration in the roots of *Pinus ponderosa* was higher in arid places. Some studies believed that this may be because plants fail to use stored carbon under drought conditions, while other studies suggested photoassimilates were allocated actively to storage at the expense of growth [11]. However, whether carbon storage is an active or a passive process, the decrease in growth and carbon accumulation is a beneficial process that may prolong survival under drought [19,20,48].

In this study, correlation analysis showed that stem diameter, root and stem dry weight, total dry weight, and root: shoot ratio were positively correlated with soluble sugar in leaves, starch and NSC content in stems, whereas seedling growth was not correlated with NSC content in roots. Seedling growth was not related to NSC content in taproots. This indicates that under drought stress, carbohydrates in stems and leaves may be allocated to the roots to promote root structure and growth [54]. Meanwhile, NSC accumulated in roots, and starch and NSC content in roots are beneficial for plants to resist adversity under drought stress. Some studies also have found that the relationship between root elongation and C availability is uncoupled under water deficit [17]. This is consistent with our results. This suggests that growth limitation under drought stress is not due to C availability, but is regulated or controlled by other mechanisms, such as cell wall rheology or water fluxes to growing cells, which need to be further studied [42,55]. This may be a root system adaptive strategy to drought stress. Overall, more root biomass and carbon storage in roots are beneficial to the survival of deciduous plants under drought stress and may aid recovery after stress.

## 5. Conclusions

This study investigated growth, biomass allocation, and NSC dynamics in different organs of Mongolian oak seedlings under different drought intensities and durations. The results showed that with a decrease in soil moisture content, growth and the increment of biomass in different organs decreased, but root: shoot ratio and root biomass increased. Meanwhile, under drought stress, the NSC content in the roots increased at the end of moderate drought and the whole severe drought period, making root systems an important

carbon sink. Additionally, the reduction of seedling growth was not accompanied by a decrease in NSC content in different organs under drought stress, indicating the decline in growth is not limited by carbon storage. Our results revealed adaptive mechanisms of growth and NSC allocation strategies of Mongolian oak seedlings under drought stress. Under severe drought, there were no dead Mongolian oak seedlings, indicating that the species has strong adaptability to drought, and is suitable for afforestation.

**Author Contributions:** Y.W. and X.L. conceived and designed the experiments; Y.W., S.M. and H.Z. performed the experiments; Y.W. analyzed or interpreted the data for the work; Y.W. wrote the manuscript and X.L., X.H. and W.A. revised it. All authors have read and agreed to the published version of the manuscript.

**Funding:** This research was funded by the National Key Research and Development Program of China (Grant No. 2021YFD2200302-3).

**Data Availability Statement:** Data is available upon request from the corresponding author.

**Acknowledgments:** We would also like to thank Shuai Chen and Luzhang Liu for their help with the fieldwork. We would also like to acknowledge Xinming Liang for her laboratory assistance.

**Conflicts of Interest:** The authors declare no conflict of interest.

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
