# Peer review of "Non-Structural Carbohydrates and Growth Adaptation Strategies of Quercus mongolica Fisch. ex Ledeb. Seedlings under Drought Stress"

_forests, doi:10.3390/f14020404_

Round 1

Reviewer 1 Report (Previous Reviewer 2)

I appreciate the authors regarding this content on drought and SRC allocations. It might provide good references for oak tree study, afforestation, and climate change research in the near future. However, I believe this manuscript should be revised and re-checked. I added several comments to improve the manuscript. Please find the attachment.

Regards,

Author Response

Dear reviewer:

    Thank you for the constructive comments. We provide a point-by-point response. Please see the attachment.

   Thanks for your attention and patience.

Reviewer 2 Report (New Reviewer)

The comments are attached herewith

Author Response

Dear Reviewer:

    Thank you for the constructive suggestion. We provide a point-by-point response. Please see the attachment.

    Thanks for your attention and patience!

Reviewer 3 Report (New Reviewer)

General comments

I have read the manuscript (forests -2167499). Entitle: Survival and growth adaptation strategies of Quercus mongolica seedlings under drought stress written by Yu Wang et. al., for publication of forests MDPI. In this study, the author investigated the seedling growth and NSC dynamics of Mongolian oak under different drought intensities. The author mainly found seedling height, ground diameter, total leaf area, total root length, and biomass gradually decreased as the soil moisture decreased, but the root-shoot ratio and root biomass allocation ratios gradually increased.

The overall research is well conducted, and research is obvious application potential for the readers because this research provided the information of the root system becomes a vital carbon sink under drought stress and that is beneficial for seedling survival. In this sense, this manuscript is much more valuable. However, I found a lack of story connection and some lack of potential references (some I suggested below). Overall after I evaluate and request the author for this manuscript as a “MAJOR REVISION”.

 Major suggestions

1) Abstract: Abstract should be short and concise; it should be present only important results only not need to indicate all findings. Please make it necessary revise. 

2) Introduction: The introduction is well starting with the global drought effect however author also should be covering the drought “reduction of plant morphology (reduced leaf size and stem length, leaf length/width, and vegetative growth) and physiological traits (reduction of photosynthesis, leaf water potential, and sap movement)” referring references (1) https://doi.org/10.1016/j.foreco.2020.118099 (2) https://doi.org/10.1093/treephys/tpy153

3) Hypothesis and objectives in the introduction: The author well presented the objective of the studies, in Ln. 89 to 97 but the hypothesis of the study is not clearly presented in the introduction. Author should be well-connect these two parts. The hypothesis should be very clear because, without appropriate literature, questions, or hypotheses in the introduction section the entire text will be unclear. 

Some others comment

4) Line 126 (Study area): Author should present the instrumental details (brand/company and detail address) in the materials and methods section see the Ln. 126 photosynthetic apparatus (Li-6400X, USA) and other lines (where used the instrumentation). Please make this necessary revised.

5) Line 211 & 237 (results): Author should well describe the results, but author should little bit concise the text because each figure and tables also describe the result itself. In scientific writing does not need to describe too detail.

6) Discussion (Line 297): Improve discussion more logically with include the effect of drought from a biochemical perspective. Generally, drought releases ROS (why ROS emerging in stress conditions?). Refer to articles (1) https://doi.org/10.1038/s41598-019-55889(2) https://doi.org/10.1016/j.scitotenv.2021.146466 “Under drought plant produces the ROS and plant produce antioxidant, flavonoids, and secondary metabolites play to the role for protecting the plant for detoxifying ROS and protect the plant against the stress condition by stabilizing the protein and amino acid”.

7) Discussion (Line 298-311): In discussion, author should be focus for the reference. The line 298- 311 I did not see the references. Author should be references these as well as concisely present the text.

8)Discussion (Ln. 312): Author should revise 4.1 some logically, that reduction of photosynthesis under mild and severe drought due to multiple cause such as reduced leaf size, reduction of Chl., reduce the plant water status (sap movement or leaf water potentials) that result of reduced carbon gain under drought. Refer these articles for references. articles 1) DOI 10.1007/s10342-016-1011-6 and 2) DOI: 10.1016/j.scienta.2018.11.021

9) Line 428 (conclusion): The outlook and conclusion should not be repetitive in the abstract or a summary of the results section. I would love to read striking points and take-home messages that will linger in the readers’ minds. What is the novelty, how does the study elucidate some questions in this field, and the contributions the paper may offer to the scientific community?

10) Line 454 (References): please double-check the citations, their style, spell check, and other grammatical errors. moreover, the author should cut the old and less matching literature and include the latest literature some of them are above.

Good Luck!

Author Response

Dear Reviewer:

    Thank you for the constructive suggestion. We provide a point-by-point response. Please see the attachment.

    Thanks for your attention and patience!

Round 2

Reviewer 1 Report (Previous Reviewer 2)

I appreciate authors tried to improve the quality of the manuscript for readers and reviewers. There are somewhat limitations on the novelty of the submitted study regarding that it is generally reported that woody plants tend to store NSCs in the roots under drought conditions; there are some limitations regarding experimental design such as comparison study with other afforestation species to support some scientific case study (i.e., NSCs variation under drought in evergreen tree species is greater than that in deciduous tree species), but it showed some pot-scale case study of Quercus mongolica seedlings. 

Nevertheless, I would like to recommend improving some discussion and statements of all parts. I'd like to recommend authors underwent English editing services for a better manuscript as well. 

There are some specific ideas and opinions of mine in the attachment.

Regards,

Author Response

Dear reviewer:

    Thank you for the constructive comments on our manuscript. We provide a point-by-point response  in the attachment. Please see the attachment.

    Kind regards.

Reviewer 2 Report (New Reviewer)

The comments are attached herewith

Author Response

Dear reviewer:

    Thank you for the constructive comments on our manuscript. We provide a point-by-point response  in the attachment. Please see the attachment.

    Kind regards.

Reviewer 3 Report (New Reviewer)

Dear Author

I have read the revised manuscript (Forests-2167499). Titled: Growth and survival adaptation strategies of Quercus mongolica seedling under drought stress for publication in forests. This is the second submission made by the author. The author addressed all the questions and suggestions that I raised the issue in the review of the original manuscript. I satisfy the author’s revisions throughout the paper. The author well-addresses all the questions and quarries in this manuscript. Especially the author improved the introduction and discussion section very well inflow. Now, this manuscript improved the flow of writing, which was comparatively shallow in the original version but in this revised copy author very well addressed all the quarries and suggestions. If there is anything needed to be revised by the author, especially English grammar, or spell check, I request this manuscript is currently in “Minor Revision” and the author may correct any further grammatical errors (if any) the author may improve in this stage. Thank you.

Author Response

Dear reviewer:

    Thank you for the constructive comments on our manuscript. We provide a point-by-point response  in the attachment. Please see the attachment.

    Kind regards.

This manuscript is a resubmission of an earlier submission. The following is a list of the peer review reports and author responses from that submission.

Round 1

Reviewer 1 Report

Please see annex

Author Response

        Thank you for the constructive comments. We have carefully considered and addressed all comments. Please see the attachment.

Reviewer 2 Report

This article investigated how different irrigation regime for Mongolian oak affects drought stress responses over time. Generally, this research can contribute to tree ecosystem services and key species selection for afforestation in arid/semi-arid areas. However, I would like to suggest that it is required to improve some methodology parts and discussion parts for a better quality of the manuscript. 

In this article, for instance, the soluble sugar content in each organ of Mongolian oak under W2 and W3 was higher than that under W1 at the beginning of the treatment, but decreased over time.

In contrast to previous research (e.g., Populus tremuloides Michx studies), the soluble sugar content in the leaf (i.e., not root or other parts) was higher in drought conditions (especially in W3). Please find more case studies regarding contrast results/discussion and describe some opinions about them in the Discussion part. In addition to this, it would be good to suggest what kind of irrigation regime or amount of water in terms of FC can be suggested for drought resistance. I wrote details in the manuscript. Please find the attachment. Thank you.

Author Response

(The authors gave the same response as above.)

Round 2

Reviewer 1 Report

The major criticism made in the first round was that the article was written in a confuse way. Although the authors have made some corrections, the major drawback after the corrections is still that the article is very confuse.

In the second version, Figures 1 and 4 have been changed. I have no major objection to the change of Figure 1, which is not very helpful, but the trouble is in Figure 4, where new results of statistics have been added creating more confusion in a figure that was already complicated.

In addition, the following sentences still contain errors or contradictions:

1) At 60 d, the soluble sugar content in leaves was still significantly higher (12.62%) under W3, but lower (13.69%) under W2, compared to that under W1.

2)       NSC contents in stems under W2 and W3 were also significantly higher than those under W1 at 15 d and 60 d (P < 0.05).

3)       The soluble sugar content in taproot under W2 was significantly lower than that under W1 at 30 d and 45 d,

(at 45 is Ok, but not at 30)

The discussion is still confuse.

Author Response

Dear Reviewer:

Thank you very much for giving me the opportunity to revise the manuscript again. Please see the attachment. 

Reviewer 2 Report

First of all, I appreciate the author reflected my comments by improving many parts of a manuscript. This paper covers how sugars and starch in  Quercus can be allocated over time during the drought, and gave implications for relevant afforestation tactics or a case study of fundamental tree physiological studies. Aside from it, I also wrote a few minor comments in the attachment. Please confirm it.  Thank you.

Author Response

(The authors gave the same response as above.)
